# Presenilin: A Multi-Functional Molecule in the Pathogenesis of Alzheimer’s Disease and Other Neurodegenerative Diseases

**DOI:** 10.3390/ijms25031757

**Published:** 2024-02-01

**Authors:** Yang Sun, Sadequl Islam, Makoto Michikawa, Kun Zou

**Affiliations:** 1Department of Biochemistry, Graduate School of Medical Sciences, Nagoya City University, Nagoya 467-8601, Japan; sunyang8610@yahoo.co.jp (Y.S.); sadequl.ru40@gmail.com (S.I.); 2Department of Geriatric Medicine, School of Life Dentistry at Niigata, The Nippon Dental University, Niigata 951-8580, Japan; mmichi@ngt.ndu.ac.jp

**Keywords:** presenilin, Alzheimer’s disease, neurodegenerative diseases, trafficking, ApoE, Aβ42-to-Aβ40-converting activity of ACE, PD, FTD, HD, ALS

## Abstract

Presenilin, a transmembrane protein primarily known for its role in Alzheimer’s disease (AD) as part of the γ-secretase complex, has garnered increased attention due to its multifaceted functions in various cellular processes. Recent investigations have unveiled a plethora of functions beyond its amyloidogenic role. This review aims to provide a comprehensive overview of presenilin’s diverse roles in AD and other neurodegenerative disorders. It includes a summary of well-known substrates of presenilin, such as its involvement in amyloid precursor protein (APP) processing and Notch signaling, along with other functions. Additionally, it highlights newly discovered functions, such as trafficking function, regulation of ferritin expression, apolipoprotein E (ApoE) secretion, the interaction of ApoE and presenilin, and the Aβ42-to-Aβ40-converting activity of ACE. This updated perspective underscores the evolving landscape of presenilin research, emphasizing its broader impact beyond established pathways. The incorporation of these novel findings accentuates the dynamic nature of presenilin’s involvement in cellular processes, further advancing our comprehension of its multifaceted roles in neurodegenerative disorders. By synthesizing evidence from a range of studies, this review sheds light on the intricate web of presenilin functions and their implications in health and disease.

## 1. Introduction

Presenilin stands as a pivotal genetic player in the intricate landscape of Alzheimer’s disease (AD), particularly in its early-onset familial form. Gene mutations in presenilin have been remarkably diverse, with over 200 different forms identified, implicated in approximately 90% of familial AD pedigrees [1]. Presenilin’s primary involvement in AD stems from its role as the catalytic core of the γ-secretase complex, a transmembrane protein with nine membrane-spanning domains [2]. Functioning as an aspartyl protease, presenilin, with aspartic acid at its active center, cleaves the amyloid β protein (Aβ) at the γ position of its substrate, the amyloid precursor protein (APP). This process leads to the generation of Aβ peptides, a hallmark feature in AD pathology [3].

In this review, we aim to deepen the understanding of presenilin and provide new explanations for neurodegenerative diseases. Going beyond traditional perspectives, we emphasize the wide-ranging roles of presenilin in cell biology, offering a fresh perspective on unraveling the mechanisms of neurodegenerative diseases. We highlight the importance of exploring the multifaceted functions of presenilin in driving field advancement and future research, providing profound insights for upcoming studies. Additionally, we further delve into the newly discovered functions of presenilin, discussing the close associations between these functions and known cellular physiology and pathological processes. Detailed explanations of how these associations impact the mechanisms of neurodegenerative diseases are provided, emphasizing the regulatory role of presenilin at various cellular levels. This in-depth interpretation not only underscores the diversity of presenilin but also offers profound insights for future research, potentially paving the way for new directions in the treatment strategies for neurodegenerative diseases.

## 2. Structure

Presenilin is a polytopic transmembrane protein encoded by two homologous genes, *PSEN1* and *PSEN2*. *PSEN1* is located on chromosome 14, while *PSEN2* is on chromosome 1. These genes encode presenilin proteins, which share structural similarities. Presenilin has nine transmembrane domains with both N- and C-termini located in the cytoplasm [4]. Presenilin undergoes post-translational modifications, including endoproteolysis, to generate a 30 kDa N-terminal fragment (NTF) and a 20 kDa C-terminal fragment (CTF) [5]. These two fragments form the γ-secretase complex (as discussed below), and it has been established that the aspartic acid residues present in them possess functional activity, acting as intramembrane cleaving proteases in the hydrophobic environment of the membrane [6]. Subsequent studies revealed that γ-secretase, which cleaves the APP protein at the γ position to produce Aβ, exists as a complex of four proteins, including presenilin [7]. These include Nicastrin, PEN-2, APH-1, and presenilin [1,8]. Nicastrin and APH-1 first form a subcomplex in the endoplasmic reticulum (ER), which then binds to the subcomplex formed by presenilin and PEN-2. Additionally, APH-1 has been found to directly bind to the C-terminal end of presenilin, shedding light on the molecular mechanism of how the four components of the γ-secretase complex cleave Aβ from APP [8,9,10].

## 3. Association with Alzheimer’s Disease

For an extended period, presenilin has played a pivotal role in Alzheimer’s disease research, particularly as the core of the γ-secretase complex. Through in-depth exploration of presenilin’s functionality, we underscore its critical role in traditional Alzheimer’s disease research while also highlighting its extensive involvement in cell biology. We comprehensively summarize presenilin’s functions in various aspects, including calcium signaling, synaptic plasticity, apoptosis, and the Notch signaling pathway, revealing the multifaceted nature of presenilin (Figure 1). This comprehensive observation provides us with a novel and profound perspective on the mechanisms underlying neurodegenerative diseases.

### 3.1. Presenilin and APP Processing

Presenilin mutations are associated with a distinctive phenotype of AD, characterized by an early onset, often occurring in the fourth to fifth decades of life. Mutated presenilin proteins, especially *PSEN1* mutants, disrupt the normal processing of APP by the γ-secretase complex, favoring the production of Aβ42 over Aβ40. This alteration in the Aβ42/Aβ40 ratio is particularly significant because Aβ42 has a higher tendency to aggregate and form toxic oligomers and fibrils in the brain, leading to the formation of amyloid plaque—a pathological hallmark of AD [11,12,13,14]. The hastened aggregation of Aβ, stemming from presenilin mutations, accelerates early cognitive decline and neuronal dysfunction. These noxious Aβ species not only impair synaptic function but also instigate neuroinflammation, ultimately leading to neuronal death.

In addition, presenilin interacts with a network of proteins that modulate its functions and contribute to AD pathology. One such protein is nicastrin, a subunit of the γ-secretase complex. Nicastrin stabilizes presenilin and promotes its proper maturation, thus influencing γ-secretase activity [15]. Moreover, nicastrin mutations can disrupt γ-secretase function and contribute to Aβ accumulation in AD [16]. Another critical player is PEN-2, which forms a stable complex with presenilin and is required for γ-secretase activity [17]. PEN-2 mutations can impair this interaction, leading to the onset of familiar AD [18,19]. Understanding the intricate interplay between presenilin and its associated proteins is vital for unraveling the complexities of AD pathogenesis.

### 3.2. Notch Signaling and Other Substrates of Presenilin

Beyond its role in Aβ production, presenilin plays a crucial role in the Notch signaling pathway. Notch signaling is crucial for cell fate determination, development, and tissue homeostasis. Presenilin-mediated cleavage releases the intracellular domain of Notch (NICD), allowing it to translocate to the nucleus and regulate gene expression [20,21]. Disruption of Notch signaling due to presenilin mutations can lead to familial AD [22,23] and has been implicated in certain cancers [17]. This altered processing disrupts normal cell activities, impacting neuronal survival [24,25]. Understanding these interactions is vital for developing targeted therapies to modulate γ-secretase and restore normal Notch signaling in AD [1].

Additionally, presenilin has several other substrates, including ErbB4, a tyrosine kinase receptor for neuregulins. Study suggests that ErbB4 may mediate a novel signaling function independent of its canonical role as a tyrosine kinase receptor [26]. Another substrate, CD44, undergoes presenilin-dependent intramembrane proteolysis, leading to the liberation of its intracellular domain and the secretion of an Aβ-like peptide [27].

Presenilin 1 plays a crucial role in the maturation and trafficking of N-cadherin to the plasma membrane [28]. Notably, presenilin 1 mutations disrupt the production of the N-cadherin intracellular fragment, leading to a failure in suppressing CREB-dependent transcription [29]. These findings suggest an alternative explanation for FAD that is separate from the widely accepted “amyloid hypothesis”: dysfunction in transcription regulatory mechanisms. Furthermore, a total of 149 γ-secretase substrates have been identified to date [30,31,32]. These substrates represent a valuable resource that may facilitate the future development of drugs inhibiting or modulating γ-secretase activity in a substrate-specific manner.

### 3.3. Presenilin and Synaptic Dysfunction and Neuronal Loss

Synaptic dysfunction is an early event in AD pathogenesis, leading to cognitive decline. Presenilin mutations disrupt calcium homeostasis, impair neurotransmitter release, and compromise synaptic plasticity [33]. This dysregulation can result from altered store-operated calcium entry (SOCE) mechanisms, which play a critical role in maintaining neuronal function. A study by Cheung et al. [34] demonstrated that mutant presenilin disrupts SOCE, leading to abnormal calcium signaling in neurons. This disruption can trigger excitotoxicity, mitochondrial dysfunction, and ultimately neuronal death. Another study has also demonstrated that presenilin interacts with synaptic proteins, including PSD-95, critical for synaptic structure and function [35]. These alterations in synaptic function contribute to memory deficits and cognitive impairment in AD patients. Therapeutic strategies aimed at preserving synaptic integrity by targeting presenilin-related mechanisms hold promise for AD treatment.

Emerging evidence suggests that presenilin also plays a role in neuroinflammation, a key feature of AD pathology. Mutant presenilin can activate microglia and astrocytes, leading to the release of proinflammatory cytokines and chemokines [36,37]. This chronic inflammatory response exacerbates neuronal damage and accelerates disease progression. Targeting presenilin-related neuroinflammation represents a potential avenue for AD therapeutics.

AD is characterized not only by Aβ pathology but also by widespread neuronal network dysfunction. Presenilin mutations have been linked to aberrant network activity in the brain. Research by Palop et al. [38] demonstrated that transgenic mice expressing mutant presenilin exhibit hippocampal hyperexcitability and epileptic seizures, suggesting a role for presenilin in regulating neuronal network activity. These findings underscore the far-reaching consequences of presenilin dysfunction beyond Aβ production.

Neuronal loss is a hallmark of advanced AD. Dysregulated Aβ production, neuroinflammation, and synaptic dysfunction driven by presenilin mutations all contribute to progressive neurodegeneration [39]. Understanding the molecular mechanisms underlying presenilin-mediated neuronal loss is crucial for developing interventions that can preserve neuronal function and slow disease progression.

Presenilin’s involvement in apoptosis, or programmed cell death, is another causative factor for neuronal loss. Dysregulated apoptosis is a hallmark of neurodegenerative diseases, including AD. Studies have shown that increased sensitivity to apoptosis in neural cells expressing mutant presenilin-1 is linked to perturbed calcium homeostasis and enhanced oxyradical production [40,41,42]. Another study demonstrated that mutant PS-2 not only induced p53 expression but also led to an increase in miR-34a expression. This suggests that mutant PS-2 may contribute to the apoptosis of neuronal cells by activating the p53/miR-34a axis [43]. This apoptotic vulnerability contributes to the progressive loss of neurons seen in AD.

Oxidative stress, resulting from an imbalance between free radicals and antioxidant, increased oxidative stress in neurons [44]. This oxidative damage can lead to lipid peroxidation, protein oxidation, and DNA damage, contributing to neurodegeneration in AD [42]. A study revealed that oxidative stress enhances PS1 protein levels in lipid rafts via up-regulation of PS1 transcription, which may constitute the mechanism underlying the oxidative stress-associated promotion of Aβ production [45]. In addition, our latest study indicates that dysfunction of presenilin may reduce intracellular ferritin levels and is involved in AD pathogenesis through increasing susceptibility to oxidative damage [46].

The blood–brain barrier (BBB) is essential for maintaining the brain’s microenvironment. Recent studies have revealed that presenilin is involved in BBB integrity through interactions with a reduced barrier function, reduced drug efflux pump activity, and diminished glucose metabolism [47,48] Dysfunction of the BBB can lead to increased neuroinflammation and infiltration of peripheral immune cells, exacerbating AD pathology [49].

Nerve growth factor (NGF) and brain-derived neurotrophic factor (BDNF) are crucial molecules in the nervous system, essential for the survival, development, and maintenance of neurons. They engage in cell signaling by binding to cell surface receptors, facilitating neuronal growth, differentiation, and survival.

### 3.4. Presenilin and Organelle Dysfunction

Endoplasmic reticulum (ER) stress is a cellular response to the accumulation of misfolded proteins in the ER, and it has been implicated in AD pathogenesis. Presenilin’s role in ER calcium regulation and protein folding quality control positions it as a key player in ER stress [50]. A study by Stutzmann et al. [51] demonstrated that presenilin mutations disrupt ER calcium homeostasis, leading to ER stress and subsequent neuronal dysfunction. This link between presenilin and ER stress highlights its multifaceted impact on cellular physiology.

Mitochondrial dysfunction is a common feature of neurodegenerative diseases, including AD. Recent studies have shown that presenilin directly interacts with mitochondria and affects mitochondrial dynamics [52]. Dysfunctional mitochondria can lead to oxidative stress and energy deficits, contributing to neuronal damage.

Autophagy, a critical mechanism for clearing damaged organelles and protein aggregates, is also regulated by presenilin. Emerging research suggests that presenilin influences autophagy through interactions with proteins such as Beclin-1 [53]. Dysregulated autophagy can lead to the accumulation of toxic protein aggregates, further exacerbating neurodegeneration in AD.

### 3.5. Presenilin and Tau Pathology

In addition to Aβ, the formation of neurofibrillary tangles, caused by the accumulation of hyperphosphorylated tau protein, is a pathological hallmark of AD. Research by Shipton et al. [54] demonstrated that tau protein is required for Aβ to impair synaptic plasticity in the hippocampus and suggested that the Aβ-induced impairment of LTP is mediated by tau phosphorylation. Many studies demonstrated that presenilin mutations exacerbate tau hyperphosphorylation and aggregation in mouse models [55,56,57]. Moreover, another study revealed that loss of presenilin function enhances tau phosphorylation and aggregation in mice [58]. This interaction between presenilin and tau pathology further highlights molecular events contributing to AD pathogenesis.

## 4. Novel Functions of Presenilin

In delving into newly discovered functions, we elaborate on multiple novel aspects of presenilin, such as its role in cellular transport, regulation of APOE secretion, and interaction with ApoE (Ref. Figure 1). Emphasis is placed on the close association of these new functions with neurodegenerative diseases, offering profound insights. This multifaceted research not only enhances our comprehensive understanding of presenilin’s diverse functions but also opens up rich possibilities for the treatment and prevention of neurodegenerative diseases.

### 4.1. Presenilin and Trafficking Function

Since 2000, presenilin, particularly presenilin 1, has been found to have the function of trafficking and turnover of various type I transmembrane proteins. Interestingly, this trafficking function of presenilin is both selective and bidirectional [1,59]. For instance, presenilin-deficient cells or cells introduced with presenilin mutations or γ-secretase inhibitors significantly inhibit the maturation and surface localization of TrkB, nicastrin, N-cadherin, and ApoER2 [60,61,62]. The immature form of nicastrin protein accumulates in the ER, and presenilin is suggested to play a role in trafficking membrane proteins from the ER to the Golgi and cell surface. On the other hand, these presenilin function inhibitions conversely increase the maturation and surface localization of APP, integrin β1, telencephalin, EGFR, and TREM2 [59,63,64,65]. Notably, the immature protein of integrin β1 in the ER is significantly reduced, while mature integrin β1 on the cell surface is increased, suggesting that presenilin may also suppress the trafficking of membrane proteins from the ER to the Golgi and cell surface [59]. It can be anticipated that presenilin is deeply involved in the metabolism and distribution of these substrate membrane proteins, which function as receptors for neuronal signaling, adhesion, differentiation, and growth. However, the mechanism by which presenilin’s trafficking function is involved in the molecular pathogenesis of AD (Aβ deposition, tau phosphorylation) remains unknown.

### 4.2. Presenilin and ApoE

Recently, we discovered that presenilin controls the secretion and intracellular localization of ApoE. The ε4 allele of the ApoE gene accounts for over 90% of sporadic AD cases and is known to accelerate the onset of AD by reducing Aβ clearance ability. In presenilin 1,2-deficient cells, ApoE secretion is abolished, and remarkably, the intracellular localization of ApoE shifts from the cytoplasm to the cell nucleus (Figure 2A–C). A decrease in ApoE secretion and an increase in nuclear localization were observed in DAPT-treated ApoE3 knock-in cells and astrocytes (Figure 2D–H) [66]. Presenilin has been found to control not only the secretion but also the nuclear localization of proteins such as ApoE, revealing a previously unknown role of presenilin in regulating the secretion of ApoE and suggesting its involvement in the onset of sporadic AD. Moreover, we also found that intracellular ApoE4 inhibits γ-secretase activity and thereby induces an increase in the Aβ42/40 ratio via binding to the γ-secretase complex (Figure 3). This result suggests a novel mechanism in which intracellular APOE4 contributes to the pathogenesis of SAD by inhibiting γ-secretase activity [67]. As many γ-secretase inhibitors have faced successive failures in clinical trials, our findings suggest the possibility that enhancing presenilin function rather than inhibiting it could reduce the Aβ42 ratio in the brain and increase ApoE secretion, potentially leading to the development of new therapeutic drugs.

### 4.3. Presenilin and Aβ42-to-Aβ40-Converting Activity of Angiotensin-Converting Enzyme (ACE)

Our investigations demonstrated that mouse and human brain homogenates exhibit an enzyme activity converting Aβ(1–42) to Aβ(1–40), and the major part of this converting activity is mediated by ACE, reducing the Aβ42/40 ratio [68,69]. Presenilin 1 deficiency abolished Aβ42-to-Aβ40-converting activity. Notably, presenilin mutations found in FAD impaired the Aβ-converting activity of ACE [70,71]. This intricate relationship between presenilin and ACE sheds light on their collaborative role in regulating Aβ levels, offering potential insights into therapeutic interventions for Alzheimer’s disease.

### 4.4. Presenilin and Neurotrophic Factors

The neurotrophic factors, including Brain-derived neurotrophic factors (BDNF), play a pivotal role in the growth, survival, and function of neurons [72]. In the context of AD, BDNF depletion is associated with tau phosphorylation, Aβ accumulation, neuroinflammation and neuronal apoptosis [73,74]. Many studies have shown that presenilin may intersect with the BDNF signaling pathway. A study showed that PS1-knockout neurons show defective ligand-dependent internalization and decreased ligand-induced degradation of TrkB and Eph receptors [75]. Stimulation of BDNF leads to tau dephosphorylation through activation of TrkB and phosphatidylinositol 3-kinase (PI3K) signaling [76]. Another study demonstrated that PS1 deficiency causes autophagy suppression in human NSCs via downregulating ERK/CREB signaling [77]. Upregulation of BDNF by the extracellular regulated kinases/cyclic AMP response element-binding protein (ERK/CREB) signaling pathway can ameliorate the Aβ-induced neuronal loss and dendritic atrophy [78]. *PSEN1* may also be involved in Wnt signaling by controlling β-catenin stability. *PSEN1* can promote the phosphorylation of β-catenin and inhibit cyclin D1, CDK6, and c-Myc molecules, as well as cell-cycle progression [79]. Wnt/β-catenin signaling pathways are activated in the process of BDNF-induced iPSC differentiation [80]. Furthermore, several studies have also established a direct correlation between presenilin and BDNF. A study demonstrated that the aberrant functioning of presenilin may have a negative impact on the production and release of BDNF. In comparison with the wild-type (WT) group, the expressions of synaptophysin and BDNF/Trk-B in the cerebellum were found to be reduced in the APP/PS1 group [81]. A recent study showed that presenilin 1, as a key player in a neuroprotective mechanism crucial for the formation of novel “survival complexes”, collaborates with N-methyl-D-aspartate receptors and neuroprotective factors EFNB1 and BDNF [82]. These findings have implications for the pathogenic effects of familial Alzheimer’s disease mutants and therapeutic strategies.

Another neurotrophic factor is nerve growth factor (NGF), which binds to the tropomyosin receptor kinase A (trkA) and the p75 neurotrophin receptor (p75NTR) [83]. TrkA has a high affinity for NGF. The NGF–trkA interaction activates various molecular pathways, including the phospholipase C-γ (PLCγ) [84]. *PSEN1* may impact phospholipase C (PLC) and protein kinase C (PKC) activation. In terms of *PSEN1* (and *PSEN2*) knockout, the expression of most PKC and PLC isoforms was reduced [85]. Another study showed that the surface trafficking of TrkA and p75NTR are altered in hiPSC-derived neurons that are differentiated from PSEN1 mutant FAD patients. The surface movement of TrkA molecules was less confined in *PSEN1* mutant neurites. Contrarily, the trafficking of p75NTR molecules was more confined in the FAD neurites. These results suggest that presenilin may regulate NGF via trkA and p75NTR receptor [86].

## 5. Relationship with Other Diseases

We delve into the associations between presenilin and other neurodegenerative diseases, presenting readers with a broader understanding of presenilin. Through in-depth research on its connections with diseases like Parkinson’s, frontotemporal dementia, Huntington’s, and amyotrophic lateral sclerosis, we unveil the diverse functions of presenilin in different conditions (Ref. Figure 1). This comprehensive exploration provides a fresh perspective on the unique role of presenilin in the field of neurodegenerative diseases, further highlighting the novelty and innovation of this research.

### 5.1. Presenilin and Parkinson’s Disease (PD)

The study of presenilin in other neurological disorders, such as PD, has also been explored, albeit to a lesser extent. While PD and AD are distinct neurodegenerative diseases with different clinical and pathological features, there are overlapping mechanisms and common genetic factors that may contribute to the risk of both conditions. One such genetic link involves presenilin mutations. Some studies have suggested that rare presenilin mutations, particularly those affecting *PSEN1*, could increase the risk of developing PD or cause atypical PD-like symptoms. A study by Chartier-Harlin et al. [87] reported that a *PSEN1* mutation was associated with autosomal dominant PD with typical PD pathology, indicating a potential role of presenilin in PD pathogenesis.

Presenilin’s potential involvement in PD may be related to its functions beyond Aβ production. Presenilin is a crucial component of the γ-secretase complex, which is involved in the proteolysis of various transmembrane proteins, including Notch and the amyloid precursor protein (APP). Dysregulation of Notch signaling, which relies on presenilin function, has been implicated in both AD and PD. Altered Notch signaling could influence neuronal differentiation, survival, and synaptic plasticity, all of which are relevant to PD pathogenesis [88]. Furthermore, presenilin is involved in calcium homeostasis, a process crucial for neuronal function and survival. Disruptions in calcium signaling, as observed in presenilin mutations, can lead to excitotoxicity and mitochondrial dysfunction, both of which are implicated in the pathogenesis of PD [89].

In the context of Parkinsonism, cognitive decline commonly manifests as the primary symptom in the majority of PSEN1 mutations, with Parkinsonism emerging in the later phases of the disease. However, the recent review article provides an excellent summary of examples of mutations linked to PD or dementia with Lewy bodies (DLB) as initial symptoms. Moreover, it illustrates the potential association between PSEN1 mutations and PD, encompassing abnormal protein folding clearance, neuroinflammation, endosomal dysfunction, and more. Additionally, the article discusses the interaction of PSEN1 with PD-related genes, including PRKN and PINK1, and explores how PSEN1 participates in mitochondrial pathways by influencing APP cleavage and the formation of AICD [90]. PSEN1 could impact APP cleavage, thereby controlling the formation of the APP intracellular domain (AICD). AICD could interact with FOXO3, which enhances the Pink1 expression. AICD may impact the expression of several genes involved in mitochondrial dynamics, for example, by reducing the expression of DNM1L/Drp1 and MFN2 (mitofusin 2). In addition, AICD is involved in the expression of mitophagy/autophagy markers. AICD enhances LC3-II expression but decreases the expression of SQSTM1, TIMM and TOMM. Through these genes, upregulated PINK1 could stimulate PRKN expression and mitochondrial functions. The PRKN-PSEN1-PINK1 cascade through AICD interactions could control the mitochondrial pathways (biogenesis, organelle trafficking and mitophagy) and autophagy. PSEN1 mutations could possibly result in PD or PD-like phenotypes via the impairment of PRKN-PINK1-dependent mitochondrial processes [91,92]. PD and mitochondria have been verified to be closely related, and abnormal mitochondrial pathways could play a key role in disease progression [92].

Studies on PSEN1 Leu166Pro and exon9 deletion revealed that PSEN1 may interact with alpha synuclein. This interaction may occur inside the different membrane compartments such as synaptic vesicles, Golgi apparatus or mitochondria. Mutant PSEN1 and alpha synuclein may prevent the release and appropriate transport of alpha synuclein to phagosomes and autophagosomes. PSEN1 mutations may result in a stronger interaction between PSEN1 and alpha synuclein. This interaction may inhibit the release of alpha synuclein to proteosomes or autophagosomes, leading to the aggregation of alpha synuclein. Further studies are needed to determine how a mutant PSEN1–alpha synuclein interaction may impact the formation of Lewy bodies [93].

In a study of German early-onset Alzheimer’s disease (EOAD) patients employing whole-exome sequencing, variants linked to presenilin 2 (*PSEN2*) were identified [94]. However, the study did not furnish direct evidence suggesting a definitive relationship between *PSEN2* and PD.

It is worth emphasizing that the role of presenilin in PD is still a topic of ongoing research, and the precise mechanisms by which presenilin may influence PD pathology are not fully understood. Given the complex and multifaceted nature of neurodegenerative diseases, exploring the potential links between presenilin and PD may provide valuable insights into the shared molecular pathways and genetic factors that contribute to these disorders.

### 5.2. Presenilin and Frontotemporal Dementia (FTD)

Studies indicate that mutations in the *PSEN1* gene may contribute to dysfunction in FTD through presenilin1 dysfunction, and the development of FTD may be influenced by the extent of loss of function in the *PSEN1* gene and the resulting tau pathophysiology [95,96]. In another study utilizing nested primers, PSEN1gene products with deletions within the exon 4–8 region were observed. These findings propose a potential association between alternative transcription of presenilin 1 and FTD [97]. Current research suggests a potential connection between presenilin and FTD. Specific mutations in the *PSEN1* gene, such as Leu113Pro, Gly183Val, Leu226Phe, Met233Leu, or Arg352 insertion, have been linked to the manifestation of FTD-like symptoms. These symptoms may include behavioral and language variant diseases. This study also summarized potential mechanisms through which PSEN1 is implicated in FTD, including loss-of-function mechanisms, aberrant splicing, and regulation of Tau-related pathways [90]. However, compared to Alzheimer’s disease, there is relatively less research on the relationship between presenilin and FTD. Further in-depth studies are required to comprehensively understand the specific connections and mechanisms between the two.

### 5.3. Presenilin and Huntington’s Disease (HD)

Research has indicated a potential association between presenilin and HD. HD is a neurodegenerative disorder caused by a mutation in the huntingtin gene. Presenilin’s involvement in the pathogenesis of HD may be linked to autophagy [98].

Several studies have demonstrated the presence of autophagy in the brains of patients suffering from HD [99] and in animal models of HD [100]. It is hypothesized that the lysosomal system, particularly the process of autophagy, is involved in the clearing of ubiquitinated inclusions in HD [101]. Presenilin1 is essential for v-ATPase targeting to lysosomes, lysosome acidification, and proteolysis during autophagy [102,103]. These studies suggest that the aberrant function of presenilin may lead to disturbances in the autophagosome–lysosome system, influencing the progression of HD. However, it is essential to note that further in-depth research is required to elucidate the specific roles and mechanisms of presenilin in the context of HD, providing a comprehensive understanding of its involvement in various neurodegenerative disorders.

### 5.4. Presenilin and Amyotrophic Lateral Sclerosis (ALS)

ALS is an adult-onset neurodegenerative disease characterized by the selective death of upper and lower motor neurons, ultimately leading to paralysis and death. A recent study confirmed that presenilin-1 mutations are a cause of primary lateral sclerosis-like syndrome [104]. The connection between presenilin and ALS is still under investigation, with emerging evidence suggesting potential links. Pathological changes in ALS are closely associated with pronounced and progressive changes in mitochondrial morphology [105,106]. Additionally, presenilin mutations deregulate mitochondrial Ca^2+^ homeostasis and metabolic activity, causing neurodegeneration in Caenorhabditis elegans and cultured rat hippocampal neurons [107,108]. These studies indicated that presenilin mutation may contribute to ALS by impairing mitochondrial function.

TDP-43 is a major constituent of ubiquitin-positive cytoplasmic aggregates present in neurons of patients with frontotemporal lobular dementia and ALS [109]. A study has suggested that the presenilin-binding protein ubiquilin 1 (UBQLN) serves as a polyubiquitin-TDP-43 cochaperone, facilitating the autophagosomal delivery and/or proteasome targeting of TDP-43 aggregates [110].

In a targeted sequencing analysis of 169 genes among 242 individuals of Caucasian descent in the United States, two presenilin mutations (*PSEN1* W203C and *PSEN1* I249L) were identified in individuals with amyotrophic lateral sclerosis (ALS) [111]. Another study suggested that the *PSEN1* I249L mutation may contribute to increased Aβ42 production and elevated Aβ42/Aβ40 ratios [112]. The understanding of presenilin’s role in ALS is an evolving area of research, and more studies are needed to elucidate the specific mechanisms involved.

## 6. Treatment and Research Progress

Research on Alzheimer’s disease (AD) has spurred investigations into therapeutic approaches targeting the γ-secretase complex, particularly its crucial component, presenilin. Drug development aimed at modulating γ-secretase activity for AD treatment faces challenges, as the complex is involved in various cellular processes with potential unintended consequences upon inhibition.

Several drugs, such as semagacestat and avagacestat, targeting γ-secretase, underwent clinical trials but were discontinued due to off-target effects and cognitive decline in some patients [113]. Ongoing research seeks to refine γ-secretase inhibitors for improved safety and efficacy. Alternative strategies focus on mitigating neuroinflammation and synaptic dysfunction associated with presenilin dysfunction. Anti-inflammatory drugs and immunomodulatory therapies are being explored to counteract neuroinflammation in AD [36]. Efforts to enhance synaptic plasticity and neurotransmission through pharmacological interventions are also underway [114].

Personalized medicine approaches are gaining momentum, recognizing the need for tailored treatments based on specific presenilin mutations. Gene editing technologies have been explored to correct certain presenilin mutations [115]. Recent studies have shown that CRISPR-Cas9 gene editing technology is able to selectively disrupt PSEN1 mutations leading to an autosomal dominant form of early-onset AD and counteract the AD-associated phenotype [116,117]. Despite challenges, targeting the γ-secretase complex, especially presenilin, remains a therapeutic focus in AD research. Balancing the reduction of amyloid beta production with the preservation of essential γ-secretase functions poses a significant challenge in AD drug development [118].

## 7. Conclusions

Presenilin, initially identified for its role in Alzheimer’s disease (AD), has emerged as a multifunctional protein with diverse roles in cellular physiology and pathology. Beyond its well-known involvement in Aβ production, presenilin plays a pivotal role in various cellular processes, including calcium signaling, synaptic plasticity, apoptosis, Notch signaling, autophagy, ER stress, mitochondrial function, oxidative stress, blood–brain barrier integrity, and tau pathology. In addition, we have summarized some recently discovered new functions of presenilin, including trafficking function, regulation of APOE secretion, interaction with ApoE and change of γ-secretase activity in different APOE genotypes, induction of oxidative damage through ferritin, and alteration of ACE-mediated Aβ42-to-Aβ40-converting activity.

Understanding the intricate molecular mechanisms underlying presenilin’s multifaceted roles is crucial for gaining insights into both normal cellular physiology and the pathogenesis of various diseases, including AD. The γ-secretase complex, with presenilin at its core, serves as critical molecular machinery involved in the processing of multiple transmembrane proteins, including Notch and APP. Dysregulation of γ-secretase activity due to presenilin mutations disrupts the cleavage of these substrates, leading to profound consequences in neurodegenerative diseases like AD.

While challenges remain, ongoing research into presenilin and its associated pathways offers hope for novel therapeutic interventions that can slow or even halt the progression of AD and other neurodegenerative diseases.

## 8. Method

In our research, we conducted a comprehensive literature review on the multifunctionality of presenilin in neurodegenerative disorders. Through systematic keyword searches and exploration of highly cited classical articles in databases such as PubMed and ScienceDirect, we aimed to cover recent research comprehensively. Our focus was on selecting recent publications to gain the latest insights into presenilin’s diverse functions in various neurodegenerative disorders. Emphasizing classical articles with high citations ensured a strong theoretical foundation for our review, enabling a thorough exploration of presenilin’s role. The genetic variation data were obtained from AlzForum URL (accessed on 22 January 2024) (https://www.alzforum.org/mutations/psen-1 and https://www.alzforum.org/mutations/psen-2). This literature review approach aimed to provide robust support for our study, contributing to a comprehensive understanding of presenilin’s multifunctionality in the context of neurodegenerative disorders.

## Figures and Tables

**Figure 1 ijms-25-01757-f001:**
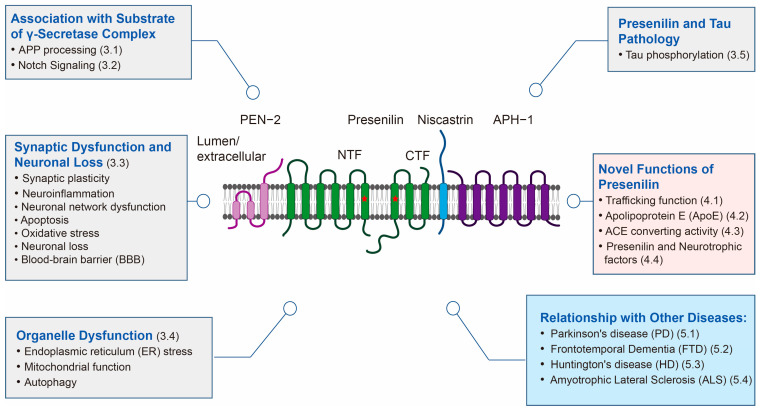
Multifaceted Roles of Presenilin in Neurodegenerative Diseases: A Comprehensive Diagram. This diagram illustrates the diverse functions of presenilin in Alzheimer’s disease (AD) and other neurodegenerative disorders. Grey boxes represent functions associated with presenilin contributing to AD through γ-secretase substrates, neuron loss, and organelle dysfunction. Pink boxes highlight novel functions of presenilin, indicating potential mechanisms. Blue boxes depict the relationships between presenilin and other degenerative diseases. In the center, the γ-secretase complex’s subunits and their membrane topologies are displayed. During complex maturation, presenilin undergoes proteolytic processing, resulting in amino-terminal fragment (NTF) and carboxy-terminal fragment (CTF). The catalytic aspartic acid residues in NTF and CTF are marked with stars. Additional subunits include Nicastrin, APH-1, and PEN-2.

**Figure 2 ijms-25-01757-f002:**
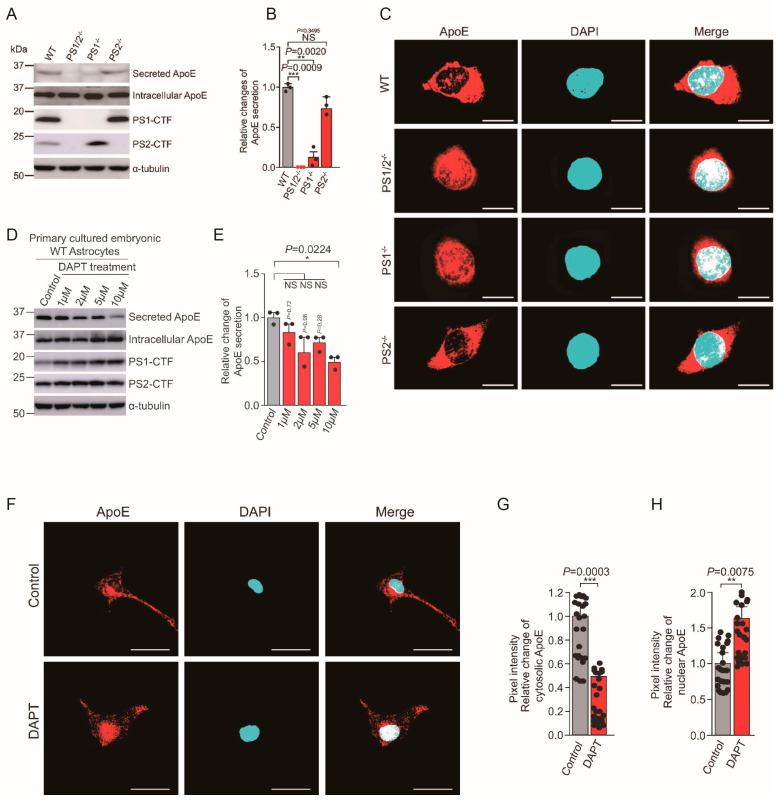
Presenilin Is Essential for ApoE Secretion, a Novel Role of Presenilin Involved in Alzheimer’s Disease Pathogenesis (Section 4.2). (**A**) Immunoblot analysis of ApoE secretion and intracellular ApoE, presenilin (PS) 1-CTF, and PS2-CTF expression in WT, PS1/2^−/−^, PS1^−/−^, and PS2^−/−^ fibroblasts cultured for 48 h in serum−free medium. (**B**) Quantification of ApoE secretion from the experiment shown in (**A**). *n* = 3. ** *p* < 0.01, *** *p* < 0.001. NS, Not significant, by one−way ANOVA followed by Tukey’s multiple−comparison tests. (**C**) Immunostaining for ApoE (red) and nuclear staining with DAPI (blue) in WT, PS1/2^−/−^, PS1^−/−^, and PS2^−/−^ fibroblasts. Scale bars, 5 μm. (**D**) Primary cultured embryonic WT astrocytes were treated with 1–10 μm of DAPT or DMSO vehicle control for 48 h in serum-free conditional medium. Levels of secreted and intracellular ApoE, PS1-CTF, and PS2-CTF were determined by immunoblotting. (**E**) Quantification of ApoE secretion from the experiment shown in (**D**). *n* = 3; * *p* < 0.05. NS, not significant, by one-way ANOVA followed by Tukey’s multiple-comparison tests. (**F**) Immunostaining for cellular distribution of ApoE (red) and DAPI staining (nuclei, blue) in primary cultured embryonic WT astrocytes treated with DAPT for 24 h. Scale bars, 50 μm. (**G**,**H**) Quantification of cytosolic and nuclear ApoE intensity from the experiment shown in F. Cytosolic ApoE was significantly decreased and nuclear ApoE was significantly increased in 10 μm of DAPT-treated astrocytes compared with control cells. *n* ≥ 24 different stained cells/group. ***p* < 0.01, ****p* < 0.001, by unpaired two-tailed *t* tests. These data are from the study by Islam et al. [66].

**Figure 3 ijms-25-01757-f003:**
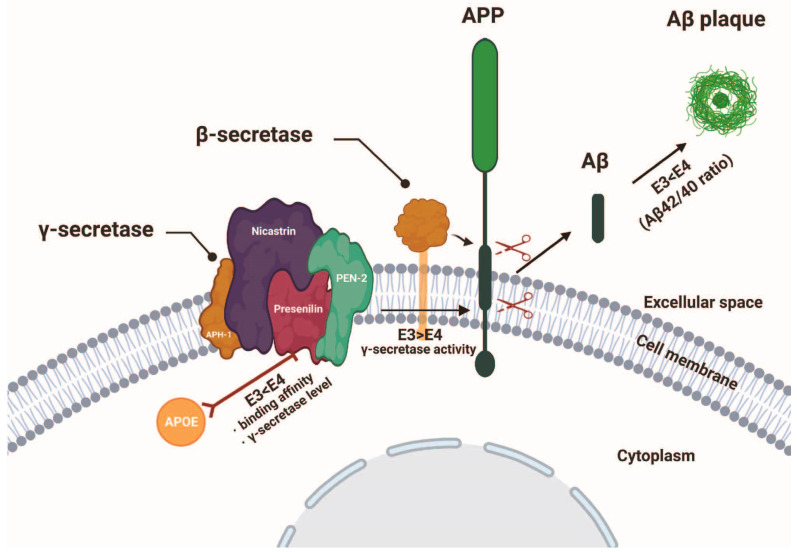
Apolipoprotein E4 inhibits γ-secretase activity via binding to the γ-secretase complex (Section 4.2). Mutations in presenilin (PS) cause familial AD (FAD) and lead to impaired γ-secretase activity and Aβ production, which results in an increased Aβ42/Aβ40 ratio. Here, we elucidated a novel regulatory mechanism for Aβ production that involves the participation of APOE in γ-secretase complex formation and activity. APOE4 inhibits γ-secretase activity and elevates Aβ42/Aβ40 ratio compared with APOE3, suggesting that APOE3 and APOE4 isoform-dependently regulate Aβ production and γ-secretase activity. Our findings provide a novel insight into the pathogenesis of AD and link together PS and APOE, the most important causative molecules in FAD and SAD, respectively. The diagram is sourced from Sun et al. [67].

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
