# Peer review of "Presenilin: A Multi-Functional Molecule in the Pathogenesis of Alzheimer’s Disease and Other Neurodegenerative Diseases"

_ijms, 2024, doi:10.3390/ijms25031757_

Round 1

Reviewer 1 Report

Comments and Suggestions for Authors In their review, the authors focus on the physiological and pathophysiological function of preselinin.
Their aim is to highlight the complexity of this therapeutic target in AD and to show that the protein also plays a role in other neurodegenerative disease.
The function of presenilin in other neurodegenerative disease is more in depth reviewed in a recent articles (Y.Y et al, MDPI, 2023) the roles of presenilin are comprehensively and up to-date summarized by the authors. The authors have a tendency to focus on their own their own data (figure 2 and figure 3), while I would expect a more balanced review better embedding the work also from other groups.
The chapter 'Presenilin with other disease seems superficial. Especially the part addressing PD. Here authors just repeat and summarize parts of the article already described above (presenilin is a component; notch signaling etc) ...
The conclusion which can taken from this part are therefore weak. Figure 1 is nice and gives a good overview over the article. Maybe the authors could add the number of the chapter to the figure. The other figures are copy-paste from other articles.

Author Response

RESPONSE TO REVIEWER1:

Re: Manuscript No. ijms-2806130.R1

We have modified blow listed formalities and made the highlighting (green and blue) in the text.

1.The function of presenilin in other neurodegenerative disease is more in depth reviewed in a recent articles (Y.Y et al, MDPI, 2023) the roles of presenilin are comprehensively and up to-date summarized by the authors. The authors have a tendency to focus on their own their own data (figure 2 and figure 3), while I would expect a more balanced review better embedding the work also from other groups. 

Response: we have cited the review (Y.Y et al, MDPI, 2023) separately in the section of “ (5.1) Presenilin and Parkinson’s disease and (5.2) Presenilin and Frontotemporal Dementia” as follows, “In the context of Parkinsonism, cognitive decline commonly manifests as the primary symptom in the majority of PSEN1 mutations, with Parkinsonism emerging in the later phases of the disease. However, the latest review article provides an excellent summary of examples of mutations linked to PD or dementia with Lewy bodies (DLB) as initial symptoms. Moreover, it illustrates the potential association between PSEN1 mutations and PD, encompassing abnormal protein folding clearance, neuroinflammation, endosomal dysfunction, and more. Additionally, the article discusses the interaction of PSEN1 with PD-related genes, including PRKN and PINK1, and explores how PSEN1 participates in mitochondrial pathways by influencing APP cleavage and the formation of AICD (93).” (page 10, lines 372-380). “Current research suggests a potential connection between presenilin and FTD. Specific mutations in the PSEN1 gene, such as Leu113Pro, Gly183Val, Leu226Phe, Met233Leu, or Arg352 insertion, have been linked to the manifestation of FTD-like symptoms. These symptoms may include behavioral and language variant diseases. This study also summarized potential mechanisms through which PSEN1 is implicated in FTD, including loss-of-function mechanisms, aberrant splicing, and regulation of Tau-related pathways (93).” (page 11 , lines 419-425). And we also have cited the works from other groups in the section of “4.1. Presenilin and Trafficking Function ” and “4.4 Presenilin and Neurotrophic factor as follows, “For instance, presenilin -deficient cells or cells introduced with presenilin mutations or γ-secretase inhibitors significantly inhibit the maturation and surface localization of TrkB, nicastrin, N-cadherin, and ApoER2 (61-63). The immature form of nicastrin protein accumulates in the ER, and presenilin is suggested to play a role in trafficking membrane proteins from the ER to the Golgi and cell surface. On the other hand, these presenilin function inhibitions have conversely increased the maturation and surface localization of APP, integrin β1, telencephalin, EGFR, and TREM2 (64-68).” (page 6, lines 239-246). “The neurotrophic factors, including Brain-derived neurotrophic factors (BDNF), play a pivotal role in the growth, survival, and function of neurons (75). In the context of AD, BDNF depletion is associated with tau phosphorylation, Aβ accumulation, neuroinflammation and neuronal apoptosis (76,77). Many studies have shown that presenilin may intersect with the BDNF signaling pathway. A study showed that PS1-knockout neurons show defective ligand-dependent internalization and decreased ligand-induced degradation of TrkB and Eph receptors (78). Stimulation of BDNF leads to tau dephosphorylation through activation of TrkB and phosphatidylinositol 3-kinase (PI3K) signaling (79). Another study demonstrated that PS1 deficiency causes autophagy suppression in human NSCs via downregulating ERK/CREB signaling (80). Upregulation of BDNF by the extracellular regulated kinases/cyclic AMP response element-binding protein (ERK/CREB) signaling pathway can ameliorate the Aβ-induced neuronal loss and dendritic atrophy (81). PSEN1 may also be involved in Wnt signaling by controlling β-catenin stability. PSEN1 can promote the phosphorylation of β-catenin and inhibit cyclin D1, CDK6, and c-Myc molecules, as well as cell-cycle progression (82). Wnt/β-catenin signaling pathways are activated in the process of BDNF-induced iPSC differentiation (83). Furthermore, several studies have also established a direct correlation between presenilin and BDNF. A study demonstrated that the aberrant functioning of presenilin may have a negative impact on the production and release of BDNF. In comparison with the wild-type (WT) group, the expressions of synaptophysin and BDNF/Trk-B in the cerebellum were found to be reduced in the APP/PS1 group. (84). A recent study showed that presenilin 1, as a key player in a neuroprotective mechanism crucial for the formation of novel "survival complexes," collaborates with N-methyl-D-aspartate receptors and neuroprotective factors EFNB1 and BDNF(85). These findings have implications for the pathogenic effects of familial Alzheimer disease mutants and therapeutic strategies. Another eurotrophic factor is nerve growth factor (NGF), which binds to the ttropomyosin receptor kinase A (trkA) and the p75 neurotrophin receptor (p75NTR)(86). TrkA has a high affinity for NGF. The NGF–trkA interaction activates various molecular pathways including the phospholipase C-γ (PLCγ) (87). PSEN1 may impact phospholipase C (PLC) and protein kinase C (PKC) activation. In terms of PSEN1 (and PSEN2) knockout, the expression of most PKC and PLC isoforms was reduced (88). Another study showed that the surface trafficking of TrkA and p75NTR are altered in hiPSC-derived neurons that are differentiated from PSEN1 mutant FAD patients. The surface movement of TrkA molecules was less confined in PSEN1 mutant neurites. Contrarily, the trafficking of p75NTR molecules was more confined in the FAD neurites. These results suggest that presenilin may regulate NGF via trkA and p75NTR receptor (89).” (page 9, lines 308-342).

2.The chapter 'Presenilin with other disease seems superficial. Especially the part addressing PD. Here authors just repeat and summarize parts of the article already described above (presenilin is a component; notch signaling etc) ...The conclusion which can taken from this part are therefore weak.

Response: We have newly cited some cases of new presenilin mutation found in PD and ALS in the section of “(5.1) Presenilin and Parkinson’s disease” and “(5.4) Presenilin and Amyotrophic Lateral Sclerosis” as follows, “In a study employing whole-exome sequencing of German early-onset Alzheimer's disease (EOAD) patients, variants linked to presenilin 2 (PSEN2) were identified (97). However, the study did not furnish direct evidence suggesting a definitive relationship between PSEN2 and PD.” (page11 , lines 403-406). “ In a targeted sequencing analysis of 169 genes among 242 individuals of Caucasian descent in the United States, two presenilin mutations (PSEN1 W203C and PSEN1 I249L) were identified in individuals with amyotrophic lateral sclerosis (ALS) (114). Another study suggested that the PSEN1 I249L mutation may contribute to increased Aβ42 production and elevated Aβ42/Aβ40 ratios (115).” (page 12, lines 459-463).

And we also have discussed the interaction of PSEN1 with PD-related genes and pathway in the “(5.1) Presenilin and Parkinson’s disease” section as follow, “In the context of Parkinsonism, cognitive decline commonly manifests as the primary symptom in the majority of PSEN1 mutations, with Parkinsonism emerging in the later phases of the disease. However, the latest review article provides an excellent summary of examples of mutations linked to PD or dementia with Lewy bodies (DLB) as initial symptoms. Moreover, it illustrates the potential association between PSEN1 mutations and PD, encompassing abnormal protein folding clearance, neuroinflammation, endosomal dysfunction, and more. Additionally, the article discusses the interaction of PSEN1 with PD-related genes, including PRKN and PINK1, and explores how PSEN1 participates in mitochondrial pathways by influencing APP cleavage and the formation of AICD (93). PSEN1 could impact APP cleavage, thereby controlling the formation of the APP intracellular domain (AICD). AICD could interact with FOXO3, which enhances the Pink1 expression. AICD may impact the expression of several genes involved in mitochondrial dynamics, for example, by reducing the expression of DNM1L/Drp1 and MFN2 (mitofusin 2). In addition, AICD is involved in the expression of mitophagy/autophagy markers. AICD enhances LC3-II expression but decreases the expression of SQSTM1, TIMM and TOMM. Through these genes, upregulated PINK1 could stimulate PRKN expression and mitochondrial functions. The PRKN-PSEN1-PINK1 cascade through AICD interactions could control the mitochondrial pathways (biogenesis, organelle trafficking and mitophagy) and autophagy. PSEN1 mutations could possibly result in PD or PD-like phenotypes via the impairment of PRKN-PINK1-dependent mitochondrial processes (94,95). PD and mitochondria have been verified to be closely related, and abnormal mitochondrial pathways could play a key role in disease progression (95). Studies on PSEN1 Leu166Pro and exon9 deletion revealed that PSEN1 may interact with alpha synuclein. This interaction may occur inside the different membrane compartments such as synaptic vesicles, Golgi apparatus or mitochondria. Mutant PSEN1 and alpha synuclein may prevent the release of the appropriate transport of alpha synuclein to phagosomes and autophagosomes. PSEN1 mutations may result in a stronger interaction between PSEN1 and alpha synuclein. This interaction may inhibit the release of alpha synuclein to proteosomes or autophagosomes, leading to the aggregation of alpha synuclein. Further studies are needed to determine how a mutant PSEN1–alpha synuclein interaction may impact the formation of Lewy bodies (96). (page10 , lines 372-402).

  1. Figure 1 is nice and gives a good overview over the article. Maybe the authors could add the number of the chapter to the figure. The other figures are copy-paste from other articles.

Response: we have added the the number of the chapter to the figure1 and figure legend 2 and 3 at the end of the title.

The attached file has already been uploaded for the manuscript revision. Please check it.

Yours Sincerely,

Kun Zou, MD PhD

Associate Professor
Department of Biochemistry
Graduate School of Medical Sciences
Nagoya City University
1 Kawasumi, Mizuho-cho
Mizuho-ku, Nagoya

Aichi 467-8601, JAPAN
Tel: (81)52-853-8141
Email: [email protected]

Reviewer 2 Report

Comments and Suggestions for Authors

In this well-done paper, the authors aimed to provide a comprehensive overview of presenilin's diverse roles in AD and other neurodegenerative disorders. It includes a summary of well-known substrates of presenilin, such as its involvement in amyloid precursor protein (APP) processing and Notch signaling, along with other functions. Additionally, the authors highlighted newly discovered functions, such as trafficking function, regulation of ferritin expression, apolipoprotein E (ApoE) secretion, the interaction of ApoE and presenilin, and the Aβ42 to Aβ40-converting activity of ACE. The authors discussed that by synthesizing evidence from a range of studies, they shed light on the intricate web of presenilin functions and their implications in health and disease.

- This narrative review is interesting, however, oddly, the authors failed to state the methods of the references’ selection.

- Furthermore, the authors should evidence the novelty of their work.

- Another limit of the paper is the absence in the paper of correlations between  presenilin, Alzheimer and NGF/BDNF.

Comments on the Quality of English Language

Minor editing of the English language required

Author Response

RESPONSE TO REVIEWER2:

Re: Manuscript No. ijms-2806130.R1

We have modified blow listed formalities and made the highlighting (yellow and blue) in the text.

1.  This narrative review is interesting, however, oddly, the authors failed to state the methods of the references’ selection.

Response: we have described the method of the references’ selection in the method section as follow, “In our research, we conducted a comprehensive literature review on the multifunctionality of Presenilin in neurodegenerative disorders. Through systematic keyword searches and exploration of highly cited classical articles in databases such as PubMed and ScienceDirect, we aimed to cover recent research comprehensively. Our focus was on selecting recent publications to gain the latest insights into presenilin's diverse functions in various neurodegenerative disorders. Emphasizing classical articles with high citations ensured a strong theoretical foundation for our review, enabling a thorough exploration of presenilin's role. The genetic variation data were obtained from AlzForum https://www.alzforum.org/mutations/psen-1 and https://www.alzforum.org/mutations/psen-2).This literature review approach aims to provide robust support for our study, contributing to a comprehensive understanding of presenilin's multifunctionality in the context of neurodegenerative disorders.” (page 13 , lines 512-523).

  1. Furthermore, the authors should evidence the novelty of their work.

Response: To evidence the novelty of our work, we have changed the section of abstract and introduction as follows, “This updated perspective underscores the evolving landscape of presenilin research, emphasizing its broader impact beyond established pathways. The incorporation of these novel findings accentuates the dynamic nature of presenilin's involvement in cellular processes, further advancing our comprehension of its multifaceted roles in neurodegenerative disorders. ” (page 1 , lines 18-21). “In this review, we aim to deepen the understanding of presenilin and provide new explanations for neurodegenerative diseases. Going beyond traditional perspectives, we emphasize the wide-ranging roles of presenilin in cell biology, offering a fresh perspective on unraveling the mechanisms of neurodegenerative diseases. We highlight the importance of exploring the multifaceted functions of presenilin in driving field advancement and future research, providing profound insights for upcoming studies. Additionally, we further delve into the newly discovered functions of presenilin, discussing the close associations between these functions and known cellular physiology and pathological processes. Detailed explanations of how these associations impact the mechanisms of neurodegenerative diseases are provided, emphasizing the regulatory role of presenilin at various cellular levels. This in-depth interpretation not only underscores the diversity of presenilin but also offers profound insights for future research, potentially paving the way for new directions in the treatment strategies for neurodegenerative diseases.” (page2 , lines 49-61).

And we have also added the summarizes in the section of “(3) Association with Alzheimer’s Disease”, “Novel Functions of Presenilin” and “Relationship with Other Diseases” as follows , “For an extended period, presenilin has played a pivotal role in Alzheimer's disease research, particularly as the core of the γ-secretase complex. Through in-depth exploration of presenilin's functionality, we underscore its critical role in traditional Alzheimer's disease research while also highlighting its extensive involvement in cell biology. We comprehensively summarize presenilin's functions in various aspects, including calcium signaling, synaptic plasticity, apoptosis, and the Notch signaling pathway, revealing the multifaceted nature of presenilin. This comprehensive observation provides us with a novel and profound perspective on the mechanisms underlying neurodegenerative diseases.” (page 3, lines 92-99). “In delving into newly discovered functions, we elaborate on multiple novel aspects of presenilin, such as its role in cellular transport, regulation of APOE secretion, and interaction with ApoE. Emphasis is placed on the close association of these new functions with neurodegenerative diseases, offering profound insights. This multifaceted research not only enhances our comprehensive understanding of presenilin's diverse functions but also opens up rich possibilities for the treatment and prevention of neurodegenerative diseases.” (page 6, lines229-235). “We delve into the associations between presenilin and other neurodegenerative diseases, presenting readers with a broader understanding of presenilin. Through in-depth research on its connections with diseases like Parkinson's, frontotemporal dementia, Huntington's, and amyotrophic lateral sclerosis, we unveil the diverse functions of presenilin in different conditions. This comprehensive exploration provides a fresh perspective on the unique role of presenilin in the field of neurodegenerative diseases, further highlighting the novelty and innovation of these research.” (page 9-10, lines 345-351).

3. Another limit of the paper is the absence in the paper of correlations between  presenilin, Alzheimer and NGF/BDNF.

Response: we have added the correlations between  presenilin, Alzheimer and NGF/BDNF in the (4.4.) Presenilin andNeurotrophic factors section as follows, “The neurotrophic factors, including Brain-derived neurotrophic factors (BDNF), play a pivotal role in the growth, survival, and function of neurons (75). In the context of AD, BDNF depletion is associated with tau phosphorylation, Aβ accumulation, neuroinflammation and neuronal apoptosis (76,77). Many studies have shown that presenilin may intersect with the BDNF signaling pathway. A study showed that PS1-knockout neurons show defective ligand-dependent internalization and decreased ligand-induced degradation of TrkB and Eph receptors (78). Stimulation of BDNF leads to tau dephosphorylation through activation of TrkB and phosphatidylinositol 3-kinase (PI3K) signaling (79). Another study demonstrated that PS1 deficiency causes autophagy suppression in human NSCs via downregulating ERK/CREB signaling (80). Upregulation of BDNF by the extracellular regulated kinases/cyclic AMP response element-binding protein (ERK/CREB) signaling pathway can ameliorate the Aβ-induced neuronal loss and dendritic atrophy (81). PSEN1 may also be involved in Wnt signaling by controlling β-catenin stability. PSEN1 can promote the phosphorylation of β-catenin and inhibit cyclin D1, CDK6, and c-Myc molecules, as well as cell-cycle progression (82). Wnt/β-catenin signaling pathways are activated in the process of BDNF-induced iPSC differentiation (83). Furthermore, several studies have also established a direct correlation between presenilin and BDNF. A study demonstrated that the aberrant functioning of presenilin may have a negative impact on the production and release of BDNF. In comparison with the wild-type (WT) group, the expressions of synaptophysin and BDNF/Trk-B in the cerebellum were found to be reduced in the APP/PS1 group. (84). A recent study showed that presenilin 1, as a key player in a neuroprotective mechanism crucial for the formation of novel "survival complexes," collaborates with N-methyl-D-aspartate receptors and neuroprotective factors EFNB1 and BDNF(85). These findings have implications for the pathogenic effects of familial Alzheimer disease mutants and therapeutic strategies. Another eurotrophic factor is nerve growth factor (NGF), which binds to the ttropomyosin receptor kinase A (trkA) and the p75 neurotrophin receptor (p75NTR)(86). TrkA has a high affinity for NGF. The NGF–trkA interaction activates various molecular pathways including the phospholipase C-γ (PLCγ) (87). PSEN1 may impact phospholipase C (PLC) and protein kinase C (PKC) activation. In terms of PSEN1 (and PSEN2) knockout, the expression of most PKC and PLC isoforms was reduced (88).Another study showed that the surface trafficking of TrkA and p75NTR are altered in hiPSC-derived neurons that are differentiated from PSEN1 mutant FAD patients. The surface movement of TrkA molecules was less confined in PSEN1 mutant neurites. Contrarily, the trafficking of p75NTR molecules was more confined in the FAD neurites. These results suggest that presenilin may regulate NGF via trkA and p75NTR receptor (89). (page 9 , lines308-342).

The attached file has already been uploaded for the manuscript revision. Please check it.

Yours Sincerely,

Kun Zou, MD PhD

Associate Professor
Department of Biochemistry
Graduate School of Medical Sciences
Nagoya City University
1 Kawasumi, Mizuho-cho
Mizuho-ku, Nagoya

Aichi 467-8601, JAPAN
Tel: (81)52-853-8141
Email: [email protected]

Round 2

Reviewer 2 Report

Comments and Suggestions for Authors

The replies to the comments I raised are satisfying.

Comments on the Quality of English Language

 Minor editing of the English language required